# Neutrophil Extracellular Trap-Driven Occlusive Diseases

**DOI:** 10.3390/cells10092208

**Published:** 2021-08-26

**Authors:** Kursat Oguz Yaykasli, Christine Schauer, Luis E. Muñoz, Aparna Mahajan, Jasmin Knopf, Georg Schett, Martin Herrmann

**Affiliations:** 1Department of Internal Medicine 3—Rheumatology and Immunology, Friedrich-Alexander-University Erlangen-Nürnberg (FAU) and Universitätsklinikum Erlangen, 91054 Erlangen, Germany; KuersatOguz.Yaykasli@uk-erlangen.de (K.O.Y.); Luis.Munoz@uk-erlangen.de (L.E.M.); Aparna.Mahajan@uk-erlangen.de (A.M.); Jasmin.Knopf@uk-erlangen.de (J.K.); georg.schett@uk-erlangen.de (G.S.); Martin.Herrmann@uk-erlangen.de (M.H.); 2Deutsches Zentrum für Immuntherapie (DZI), Friedrich-Alexander-University Erlangen-Nürnberg (FAU) and Universitätsklinikum Erlangen, 91054 Erlangen, Germany

**Keywords:** neutrophil extracellular traps, aggregation, occlusions, vessels, ducts

## Abstract

The enlightenment of the formation of neutrophil extracellular traps (NETs) as a part of the innate immune system shed new insights into the pathologies of various diseases. The initial idea that NETs are a pivotal defense structure was gradually amended due to several deleterious effects in consecutive investigations. NETs formation is now considered a double-edged sword. The harmful effects are not limited to the induction of inflammation by NETs remnants but also include occlusions caused by aggregated NETs (aggNETs). The latter carries the risk of occluding tubular structures like vessels or ducts and appear to be associated with the pathologies of various diseases. In addition to life-threatening vascular clogging, other occlusions include painful stone formation in the biliary system, the kidneys, the prostate, and the appendix. AggNETs are also prone to occlude the ductal system of exocrine glands, as seen in ocular glands, salivary glands, and others. Last, but not least, they also clog the pancreatic ducts in a murine model of neutrophilia. In this regard, elucidating the mechanism of NETs-dependent occlusions is of crucial importance for the development of new therapeutic approaches. Therefore, the purpose of this review is to address the putative mechanisms of NETs-associated occlusions in the pathogenesis of disease, as well as prospective treatment modalities.

## 1. Introduction

Neutrophils are the most abundant white blood cell type, and their function in the innate immune defense system is relatively well-defined. The formation and function of neutrophil extracellular traps (NETs) have not been elucidated in detail, although several studies related to NETs release have been conducted since the first report in 2004 [1]. NETs are web-like structures that consist of decondensed nuclear chromatin and granular proteins like neutrophil elastase (NE), cathepsin G, myeloperoxidase (MPO), and others. They are formed as the first defender of the immune system in a variety of pathological conditions. NETs offer a physical and chemical barrier against pathogens. They adhere to and clear the pathogens in the bloodstream and various ducts of the body [2]. Although NETs formation was initially thought to be a beneficial defense weapon of the host, some studies revealed deleterious effects of exaggerated NETs formation under certain conditions. The balance between the production and degradation of NETs must, therefore, be strictly regulated [3]. If the balance changes in favor of the persistence of NETs and cytotoxic proteins are not eliminated, NETs can cause tissue damage and even be involved in the development of various autoimmune diseases [4,5]. In addition, the excessive NETs formation leads to aggregated NETs (aggNETs). Despite the major advantage of aggNETs degrading inflammatory mediators, thereby fostering the resolution of inflammation, excessive NETs development carries the risk of occluding various tubular structures. It has been reported that aggNETs-driven occlusions play a role in the pathogenesis of many diseases not only in vessels but also in ducts [6,7]. Vascular occlusions containing aggNETs are found in alveolar capillaries of patients with COVID-19. The aggNETs-associated vascular occlusions induce canonical, as well as noncanonical thrombogenesis (Figure 1a) [8]. The occlusions of ducts cause pent-up secreted material, as in the case of Meibomian glands of a patient with acute blepharitis (Figure 1b) [9].

The purpose of this review is to address the factors leading to NETs formation and the potentially deleterious effects of NETs-associated occlusions in the body. Furthermore, potential prospective treatment methods will also be discussed.

## 2. Vascular Occlusions

Canonical thrombi formed after activation of the cascade of coagulation are the leading cause of vascular occlusions, including arteries, veins, and capillaries. Occluding thrombi hinder the blood flow, as they can occupy the entire lumen of the vessels. Thromboembolism occurs if a partial thrombus detaches from the vessel walls, floats in the circulation, and finally, fully occludes a distal part of the vessel. Due to its narrow relationship with inflammation, thrombosis plays a critical role in the pathophysiology of various diseases [10,11]. A number of factors can initiate thrombosis, but the role of NETs in thrombogenesis was reportedly unique [12]. Especially the elucidation of novel roles of extracellular DNA and histones in the bloodstream contributed to the reinterpretation of thrombogenesis. In 2013, the term “immunothrombosis” was coined to describe thrombi initiated by the innate immune response [13,14].

Fuchs et al. first reported the prothrombotic effect of NETs in 2010 [15]. NETs have been shown to increase thrombosis both directly and indirectly. It is still debatable whether full-size NETs, aggNETs, or the remnants of NETs are the main inducing agent. While intact aggNETs are thought to act as a scaffold for vascular occlusion, the promotion of coagulation by NETs components, including extracellular DNA via both the intrinsic and extrinsic pathways, has also been reported [16,17]. The mechanisms of NETs-induced thrombogenesis have attracted much attention and have been the subject of many investigations, as it is associated with many diseases (Figure 2). Several biomarkers, including miR-146a [18], fibrin [19], plasmin [20], Sirtuin 3 (Sirt3) [21], peptidylarginine deiminase type IV (PAD4) [22], high-mobility group box 1 protein (HMGB1) [23], and cholesterol crystal (CC) [24], have been proposed to be involved in the NETs-related noncanonical vascular occlusions that may, in addition, promote canonical thrombogenesis. Jimenez-Alcazar et al. identified a noncanonical mechanism for vascular occlusion based on employing targeted mutations of deoxyribonuclease 1 (DNase1) and DNase1l3 [25]. When the dual host protector function of these DNases against the detrimental effects of intravascular NETs is lost, NETs cannot be removed efficiently, and aggregated NETs can occlude vessels.

### 2.1. Vascular Occlusions in COVID-19

SARS-CoV-2 (severe acute respiratory syndrome coronavirus 2) continues to cause public health issues worldwide. NETs reportedly contribute to acute respiratory distress syndrome (ARDS), and fibrin-based occlusions were observed in the vasculature of patients with coronavirus disease 19 (COVID-19) [26,27]. The experimental data demonstrated a positive correlation between the quantity of NETs markers in plasma or serum from patients with COVID-19 and the severity of COVID-19. The elevated level of circulating NETs has the potential to cause and exacerbate thrombotic events. In severe COVID-19, the intravascular aggregation of NETs causes vasculopathy and can occlude the microvasculature of virtually all organs, thus precipitating organ failure and causing substantial mortality [27,28].

Indeed, severe COVID-19 is often characterized as vessel occlusions caused by intravascular NETs [8,29,30]. The lung is the first tissue affected in patients with COVID-19. The patients have been found to have higher levels of aggNETs in their lungs, blood, and tracheal aspirate fluid [31,32]. NETs are able to directly activate endothelial cells, inducing major changes that include cell death. This promotes endothelial dysfunction, fosters lung edema, and compromises the barrier function at the air/blood interphase [33]. Furthermore, the activation of the endothelium facilitates thrombosis triggered by the accumulation of NETs in the microvasculature [34]. The fact that classical antithrombotic treatments are hampered in immunothrobosis suggests that NETs are central components of vascular occlusion [25].

### 2.2. Coronary Occlusions

Cardiovascular diseases are the most common diseases with the highest death rates worldwide [11]. In myocardial infarction, atherosclerotic plaque rupture with subsequent thrombogenesis is considered the essential event in arterial occlusions, because it jeopardizes the epicardial flow. However, the fundamental mechanisms that cause coronary occlusions are still partially elusive [11,16]. Neutrophils have been shown to exacerbate vascular occlusion, and their blood concentration is considered one of the most reliable indicators for acute coronary events. However, their role in coronary thrombosis is relatively less explored compared to monocytes [35].

The discovery of NETs components in excess amounts not only in plasma and plaques from atherosclerosis patients [36] but, also, in atherosclerotic lesions in mice [37] suggests a pivotal role of NETs in atherosclerosis. This is further supported by the fact that inhibition of the DNA decondensation during NETs formation, e.g., by inhibitors of PAD4, ameliorates atherosclerosis [37]. Analyzing the compositions of thrombi is required to understand the underlying mechanisms of thrombogenesis. A substantial burden of NETs and their critical constituents were detected displaying a heterogeneous morphology in thrombectomy specimens from patients with stent thrombosis [38] and acute myocardial infarction [39]. Coronary thrombi and blood from the surrounding sites and healthy vessels were analyzed. In addition to the high amount of NETs in the thrombi, NETs components such as double-stranded DNA, NE, and MPO were detected in the vicinity of coronary thrombi but not in healthy vessels [40,41]. The associated viral infections, including COVID-19, have also been established during the pathogenesis of myocardial infarction. NETs in coronary thrombosis have shown a critical relationship to the pathogenesis of ST-elevated myocardial infarction (STEMI) [42]. Thrombin-activated platelets interact with PMNs at the site of plaque ruptures during acute STEMI, resulting in local NET formation and the delivery of active tissue factor (TF) [41]. NETs-associated TF was found significantly elevated in the coronary plasma samples obtained from patients with STEMI [43]. These findings support the hypothesis that NETs can be targeted to develop new drugs to prevent cardiac thrombosis accompanied by inflammatory conditions like viral infections. 

### 2.3. Cerebral Occlusions

Thromboembolic occlusions in the large cerebral arteries such as the internal carotid (ICA) and/or the middle cerebral artery (MCA) cause ischemic stroke, the second-leading cause of morbidity and mortality worldwide. The treatment is carried out with two strategies: (I) mechanical thrombectomy and (II) the intravenous injection of recombinant tissue plasminogen activator (tPA) [44,45]. Analyses of clots in more detail that cause ischemic stroke showed that neutrophils and NETs play pivotal roles in the pathogenesis of cerebral occlusions [44,45]. Indeed, NETs were discovered in high concentrations not only in plasma [46] but, also, in cerebral thrombi [47] taken from patients with stroke. In addition, the distribution of NETs in the thrombi from acute ischemic stroke patients has been investigated. NETs were detected in 35 out of 37 thrombi, especially in fibrin-rich areas [48]. Thus, new anti-NETs approaches aiming to develop therapies have gained momentum. In this context, treatment with DNase1 and the inhibition of PAD4 are considered good candidates [49]. 

### 2.4. Sickle Cell Disease

Sickle cell disease (SCD) is defined as the aberrant production of hemoglobin S (HbS) caused by mutations in the hemoglobin beta chain. The inability to carry adequate amounts of oxygen results from a single amino acid substitution in the gene encoding the β-globin subunit. The mutations are the hallmark of SCD [50]. Inflammation and recurrent painful vaso-occlusive crisis (VOC) are commonly seen in the course of SCD. It was previously reported that rigid red bloodcells (RBCs) obstruct the microcirculation and, thus, cause VOC. The underlying mechanisms of vaso-occlusion were understood to be more intricate, and more research is needed for clarification [50,51]. Erythrocytes are increasingly understood to be the key factor in prolonging the chemokine half-life in the circulation. They serve as dynamic cytokine reservoirs even under healthy conditions [52]. The intravascular hemolysis of abnormal hemoglobin was discovered to initiate vascular inflammation due to the release of cytokines and, further, inflammatory mediators, thus inducing excessive NETs formation during the VOC of SCD [53,54]. A recent study confirmed the parallel induction of both NETs and inflammatory agents in plasma collected during VOC compared to the steady-state conditions [55]. In order to open up new therapy options, the regulation of NETs during VOC is currently under investigation [56].

### 2.5. Effects of Nondegradable Nano- or Microparticles

Investigations of the response of neutrophils towards inorganic particles are relatively new due to the recent advances in nanotechnology. New nanotechnology-based applications have potentially negative impacts on human health caused by increased exposure to particulate matter. Consequently, developing new forms of nanoparticles is essential to reduce the potential adverse effects [57]. We have reported that neutrophils also form NETs after exposure to hydrophobic nanoparticles in a size-dependent manner. The persistence of this response may cause the formation of aggNETs. This aggregation may be important for the resolution of inflammation [58,59]. Desai and colleagues verified these findings and reported that the RIPK1-RIPK3-MLKL signaling pathways are involved [60]. Besides the aggNETs-driven resolution of inflammation, the obstructive capabilities of NETs may cause adverse effects and even fatal consequences. Indeed, the formation of occlusive co-aggregates of NETs with superparamagnetic iron oxide nanoparticles (SPION) has recently been reported to cause vascular obstructions. The coating of SPIONs with biocompatible albumin or dextran reduced NETs formation and prevented the vascular occlusion in vivo [61]. 

Since aggregation-related pathologies are prone to cause serious health problems, and even fatalities, the possible treatment methods are currently being investigated intensively. A part of the study aims to reduce inflammation by controlling neutrophils and NETs formation. For example, Feraheme^®^ (a compound used to treat iron deficiency anemia) was shown to limit the chemotactic activation after uptake by neutrophils and diminish their inflammatory properties [62].

### 2.6. Others 

In addition to their ability to bind and kill pathogens during innate immune responses, neutrophils and NETs participate in inflammatory and coagulatory responses in the circulatory system. So far, occlusions caused by NETs-associated thrombus formation have been linked to cancers [16], pre-eclampsia [63], renal ischemia-reperfusion injury [64], liver pathologies [65], and severe malaria [66,67].

## 3. Airway Occlusions

Human respiratory syncytial viruses (hRSV) are the major viral pathogens causing lower respiratory tract diseases, particularly in newborns and children under the age of five [68]. During severe RSV infection, the virus exaggerates the activities of neutrophils and eosinophils and causes extensive neutrophil accumulations in the lower airways of the lungs. This causes airway obstruction through the blockage of small airways that impedes proper breathing and leads to acute morbidity. The excessive formation of NETs is considered one of the leading causes for these small airway obstructions (Figure 2) [68,69,70]. 

In order to develop new therapeutic strategies, the mechanisms of NETs formation were studied in the context of hRSV infections. Funchal et al. revealed that the RSV Fusion protein plays a pivotal role in NETs induction via the TLR-4 or ERK/p38 MAPK pathways [71]. In another comprehensive study, RSV has been postulated to cause NETs formation via a canonical ROS-dependent mechanism [72]. The signal inhibitory receptor on leukocytes (SIRL)-1 and leukocyte-associated immunoglobulin-like receptor (LAIR)-1 have been proposed as potential targets that reduce neutrophil activity and, thus, regulate airway inflammation upon engagement of these receptors [73]. Inhaled DNase 1 treatment was proposed as an alternative therapeutic approach in NETs-induced airway obstruction during severe RSV infection [74].

## 4. Occlusions of Exocrine Glands and Ducts

### 4.1. Pancreatic Duct

The pancreatic duct, also known as the Wirsung duct, connects the pancreas and intestine through the common bile duct. Its primary function is to transport enzymes and bicarbonate, which aid digestion and neutralize the duodenal pH, respectively [75,76]. Occlusion of the pancreatic duct may cause pancreatitis. The occlusions of the ducts are directly proportional to the severity of the pancreatitis and depend on the duration of the disease [75,76].

NETs were reported to directly induce trypsin activation, inflammation, and tissue damage in severe acute pancreatitis induced by retrograde taurocholate infusion [77]. However, the direct cause of obstruction is not always found in human acute pancreatitis. Under physiological conditions, neutrophils are present in small amounts in the pancreas and enter the bicarbonate-rich pancreatic fluid and spontaneously form NETs [78]. In the case of severe inflammatory situations, the resulting neutrophilia can produce excessive amounts of NETs in the pancreas. This results in large and sticky aggregates prone to occlude pancreatic ducts. Leppkes et al. reported that sole neutrophilia in IL17 transgenic mice was the main driving force for the development of pancreatitis. In this case, excessive NETs formation was immediately induced by a high bicarbonate concentration in the pancreatic juice. Formed NETs tend to aggregate and occlude the ducts triggering focal acute pancreatic [79]. The reduced production and sizes of NETs and aggNETs in the pancreatic ducts of PAD4-KO mice prevented the development of focal pancreatitis [80,81].

### 4.2. Meibomian Gland

The Meibomian gland (MG) secretes a biological fluid (meibum) containing a significant amount and variety of lipids. Meibum is crucially important for the maintenance of a healthy ocular surface. Meibomian gland dysfunction (MGD) and the disruption of meibum homeostasis change the lipid content of the tear fluid [82,83]. The lack of lipids in the tear film promotes hyper-evaporation and tear hyperosmolarity. 

As neutrophils are the first cells to be recruited to the foci of inflammation, NETs have been discussed to drive the pathogenesis of MGD-related diseases, including dry eye disease (DED). The increased abundance of NETs in the tears and on the ocular surfaces of the patients with MGD and the positive correlation between NETs amounts and disease severity has reinforced this suspicion [84]. Although aggNETs have been known for their ability to resolve inflammation on the ocular surface [85], a recent study demonstrated that NETs are also implicated in MG terminal duct occlusions precipitating the inflammation on the ocular surface [9]. This work has provided vital new evidence to propose novel avenues in the treatment of MG occlusion-related disorders. In addition to the currently available medical treatments, such as antibiotics and nonsteroidal and steroidal anti-inflammatory drugs, NETs formation inhibitors should be considered in the therapeutic arsenal of MGD [86].

### 4.3. Periodontitis (Periodontal Crevicular Occlusions)

The gingival crevice, also called the gingival sulcus, is defined as a narrow V-shaped space between the inner aspect of the free gingival epithelium and the surrounding enamel of a tooth. Normally, its depth range is 1–3 mm. The gingival epithelium continuously produces gingival crevicular fluid (GCF) that is finally transferred into the oral cavity. It has been known for many years that the production of GCF and its composition change during inflammatory diseases like periodontitis. Therefore, GCF has been extensively studied as a diagnostic tool [87,88]. Under physiological conditions, the minimal amount of GCF present in the gingival crevice flows into the oral cavity, making the crevice a kind of duct. Upon inflammation, this fluid transforms into a purulent exudate containing large amounts of NETs and neutrophils expressing CD177 [89,90]. The exudate is extremely viscous due to the excessive amount of aggNETs. Vitkov and colleagues put forward the view of NETs-driven obstruction of the periodontal crevice. The authors speculated that the formation of a periodontal abscess might result from NETs-induced cervical obstructions [6] supported by the overproduction of NETs and the impaired clearance of NETs remnants [91]. This hypothesis has not yet been proven experimentally. However, the crucial roles of NETs in the initiation and progression of inflammatory periodontal diseases are surely worth investigating.

### 4.4. Gallstones

Obstruction of the biliary system, a common, serious, and painful condition, is one of the leading causes of hospitalization with significant morbidity and mortality. The formation of gallstones in the gallbladder or ducts, also referred to as cholelithiasis, are the most prevalent etiological event for biliary obstruction that results in biliary stasis. However, gallstone-caused biliary obstructions generate a high socioeconomic burden due to their high incidence, especially in developed countries [92,93]. Until recently, it has been proposed that gallstones simply form due to the supersaturation of cholesterol crystals. The contents of human gallstones were eventually investigated for NETs, as cholelithiasis is an occlusive condition [94]. After observing extracellular DNA and neutrophil elastase in gall sludge and gallstones, it was established that an intact NETs formation capacity is necessary to form gallstones in a murine model of cholelithiasis. Basically, NETs were crucial in the initiation and progression of gallstone formation by promoting the aggregation of biliary cholesterol and calcium crystals. Furthermore, a positive correlation between the neutrophil/lymphocyte ratio and the severity of gallstone-induced pancreatitis has been reported in clinical settings [95]. However, formal clinical trials employing NETs formation inhibition in patients with recurrent cholelithiasis are still needed [96].

### 4.5. Sialoliths

Sialoliths, also known as salivary stones, are the most prevalent obstructive disease of the salivary glands, especially for middle-aged patients. Stone formation is mostly seen in the submandibular gland, with an incidence of more than 80 percent. The parotid gland follows it with 13 percent and by sublingual and minor salivary glands at very low rates. The etiology of sialolith formation was initially considered a multifactorial interaction of calcium salts, organic and inorganic molecules, pH, and bacteria [97,98]. In two conflicting reports, inorganic materials [99] and organic components [100] have claimed to be the main components of sialoliths. However, since the role of inflammation in sialoliths formation [101] and the presence of bacterial residues, including bacterial DNA and biofilms in their structures, has been established [102], NETs have gained attention as a possible etiological factor. Furthermore, since bacterial biofilm structures at the core of sialoliths have been established, biofilms have been hypothesized to be an initial step in the formation of sialoliths [103]. Recently, neutrophil recruitment and NETs formation in the salivary system were suggested to initiate sialolithiasis. Indeed, the demonstration of NE activity associated with a high prevalence of extracellular DNA points to NETs as nidi for the formation of sialoliths [104]. The high content of bicarbonate in saliva strengthens this hypothesis as it facilitates the formation of NETs [78]. The interaction between neutrophils and the precipitated particulate matter is discussed to cause the salivary stone to grow. These results offer an alternative perspective for the “until now” proposed mechanisms of sialolithogenesis [105,106].

### 4.6. Further Obstructive Diseases

Kidney stones (nephrolithiasis) are among the oldest known diseases, affecting about 12% of the world’s population at some point in their lives [107]. Although the earliest recorded kidney stone cases date back to 4000–5000 BC, and the first medical texts for its treatment date back to 1500 BC in ancient Egypt, the etiology and appropriate treatments are still partially elusive [108]. Randall’s plaques have been widely accepted to be the nidus for kidney stone formation. Crystallization with particle retention is the most crucial event in kidney stone formation. Randall discovered calcium salt deposits in kidney stones for the first time and termed it “plaque”. These calcium deposits were later referred to as Randall’s plaques. However, although it has been more than 80 years since its first definition, not much progress has been made in the characterization of the roles of these plaques in kidney stone formation [109,110]. More than 80% of all kidney stones are based on calcium, and hypercalciuria is associated with the formation of Randall’s plaques and kidney stones [111]. Considering the evidence collected for gallstones, NETs may also drive the onset and/or progression of kidney stone formation. This is strengthened by recent studies describing the neutrophil–lymphocyte ratio as a new diagnostic marker for kidney stones and increased neutrophil activation in kidney stone formation [111,112]. 

Skin calcifications (e.g., calcinosis cutis) and stones found in large body cavities like the intestine and bladder are further candidates for NETs-driven conditions [113,114,115]. Therefore, the presence of NETs in these calcareous formations needs to be investigated.

## 5. Therapeutic Approaches to Prevent NETs-Driven Occlusions

Additionally to their protective effects against pathogens, NETs contribute to the regulation of innate and adaptive immunity. In the case of persistent neutrophil activation, the NETs form aggregates that contribute to the resolution of inflammation, reflecting the advantageous side of NETs formation. Depending on where aggNETs are formed (e.g., vessels or ducts), these large and sticky aggregates might cause various kinds of obstructions. These occlusions drive the pathophysiology and increase the severity of many diseases, including COVID-19, chole- and sialolithiasis, dry eye disease, and probably many others. Therefore, it is critical to developing new effective therapeutic strategies directed to control either the accumulation of NETs or improving their clearance at specific anatomic locations [26,116]. NETs-inducing agents are diverse, and the formation mechanisms of NETs vary depending on these triggering agents. 

Consequently, the mechanism of NETs formation has to be explored for every single condition, and then, therapeutic strategies have to be developed in a condition-specific manner. NE, MPO, PAD4, and extracellular DNA are promising targets for suppressing excessive NETs formation [11,28] and are, therefore, currently the focus of drug development (Figure 3). Among these, the inhibition of PAD4 and the treatment with DNase1 are the most promising candidates for NETs-driven occlusions [91]. Therefore, we will explore these two candidates together with others in more detail in the following paragraphs.

### 5.1. Inhibition of PAD4 Reduces Formation and Size of NETs

Peptidylarginine deiminases (PADI) are a calcium-dependent enzyme family responsible for post-transcriptional deamination/citrullination. In this process, the positively charged arginine residues are converted into uncharged citrulline residues. The PADI family consists of five members. PAD4 is unique among them. It plays a role in the formation of NETs by ensuring the decondensation of chromatin [117]. This makes PAD4 a possible therapeutic target for the treatment of occlusive NETs-related diseases. The increased activity of PAD4 and the therapeutic potential of its inhibition have already been reported in preclinical settings for the formation of thrombosis [22], acute pancreatitis [81], cholelithiasis [94], meibomian gland disfunction [9], lung injury [64], and sickle cell disease [56].

### 5.2. Deoxyribonucleases Dismantle NETs

DNases have been studied for therapeutic purposes in obstructive conditions. DNase1 cleaves DNA by breaking its phosphodiester bonds. This disrupts the structural integrity of NETs and reduces the sizes and amounts of aggNETs [118]. It has been shown that disrupting NETs with DNase1 not only prevents vascular occlusion [25] but also recanalizes the already occluded vessel [119]. Similar results have been obtained in crystal clots-driven arterial occlusion [24]. The improvement of ventilation after the inhalation of DNase1 in RSV acute bronchiolitis has also been demonstrated [71]. DNase1 is already approved as an inhalant to reduce the viscosity of the mucus in the lungs of patients with cystic fibrosis [74].

### 5.3. Inhibitors of Myeloperoxidase Reduce the Early Phases of NETs Formation

One of the most important elements of NETs, especially in the early phases, is MPO. When neutrophils encounter danger signals, MPO enters the nucleus and drives chromatin decondensation, a crucial step of NETs formation [120]. Inhibitors of MPO such as PF-1355 have been tried as NETs formation blockers in small vessel vasculitis. In this context, PF-1355 prevented excessive NETs formation and reduced leukocyte infiltration [121], making it a good candidate for further studies trying to interfere with NETs formation in occlusions. Furthermore, natural surfactants also have the ability to inhibit NETs formation in vitro [122]. Therefore, natural surfactants bear the potential to be used as therapeutical agents in occlusive conditions.

## 6. Conclusions

Neutrophils and NETs participate in the initiation, pathogenesis, and resolution phases of several inflammatory conditions. Considerable collateral damage is expected in most of the cases where neutrophils are involved. Therefore, robust regulatory mechanisms like apoptosis and NETs formation have evolved to endow neutrophils with the ability to drive the amelioration of the initial inflammation. The clearance of apoptotic neutrophils by professional phagocytes triggers potent anti-inflammatory and regenerative responses [123,124]. NETs formation and aggregation actively limit the spreading of pathogens and inflammatory mediators [7,125]. Unfortunately, those anatomical locations carrying fluids or air through the body are prone to be occluded by NETs generating heterogeneous pathologies. The study of the prevention or dissolution of such clogs will provide new therapeutic opportunities for old prevalent diseases.

## Figures and Tables

**Figure 1 cells-10-02208-f001:**
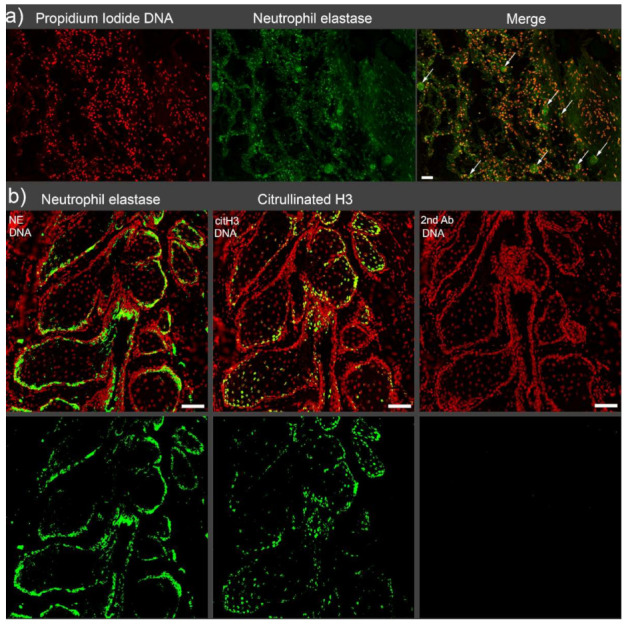
Immunofluorescence staining of AggNETs in the intravascular space and epithelial surfaces. (**a**) Neutrophil elastase (NE, green)-rich microvascular occlusions are observed in the alveolar area of a patient with deceased COVID-19. The occlusions are indicated with arrows in the merged image. Scale bar 50 µm. (**b**) Aggregated NETs (NE and citrullinated histone 3, green) fill the epithelial surfaces of stagnated acini of Meibomian glands of a patient with acute blepharitis. Scale bar 100 µm.

**Figure 2 cells-10-02208-f002:**
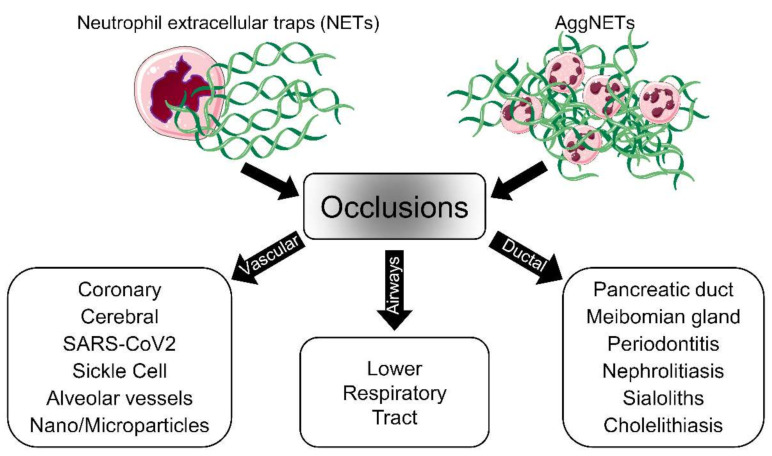
Schematic representation of aggNETs-driven pathologies. The neutrophils, as a first line of defense, eject NETs when they encounter foreign agents. Single or aggregated NETs might cause, under certain conditions, the obstruction of the flow of body fluids and, consequently, disease. The pathology depends on the site of occlusion; vascular NETs formation is associated with both canonical and noncanonical thrombogenesis in various diseases. The accumulation of AggNETs in the lower respiratory tract has been detected in cystic fibrosis patients. Exocrine gland obstruction has been observed in the pancreas and Meibomian and salivary glands. Large organs like the liver and kidneys can also be affected by the accumulation of NETs. The cartoons were modified from https://smart.servier.com accessed on 20 August 2021 in compliance with the terms of the Creative Commons Attribution 3.0 Unported License (CC BY 3.0).

**Figure 3 cells-10-02208-f003:**
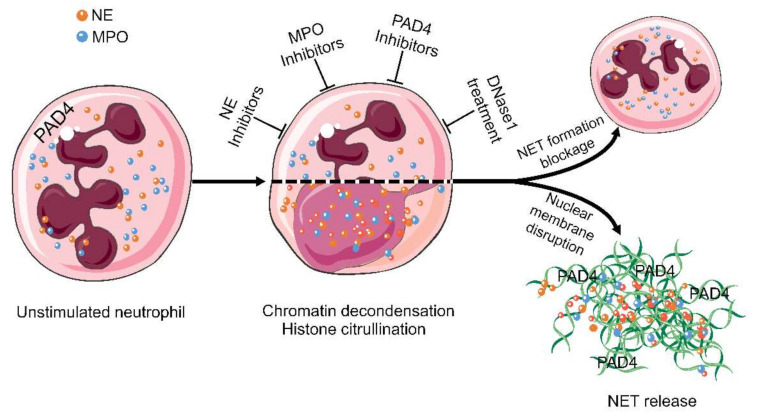
Targets in the treatment of aggNETs-driven occlusive pathologies. NE, MPO, PAD4, and extracellular DNA are promising targets for the suppression of excessive NETs formation. Enzymatic inhibitors of PAD4 and MPO have demonstrated its effectivity in preclinical settings of cholelithiasis, Meibomian gland dysfunction, and vasculitis. DNase treatment is effective at preventing death in severe sepsis and improving the symptoms in cystic fibrosis, bronchiolitis, and dry eye disease. The cartoons were modified from https://smart.servier.com accessed on 20 August 2021 in compliance with the terms of the Creative Commons Attribution 3.0 Unported License (CC BY 3.0). Abbreviations: MPO, myeloperoxidase; NE, neutrophil elastase; NET, neutrophil extracellular traps; and PAD4, peptidylarginine deiminase type IV.

## Data Availability

Primary data is available upon request.

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
