# Peer review of "Neutrophil Extracellular Trap-Driven Occlusive Diseases"

_cells, 2021, doi:10.3390/cells10092208_

Round 1

Reviewer 1 Report

In this review article, Yaykasli and colleagues have overviewed the insights into the previously and currently regarded knowledges on pathophysiological roles of NET formation. This manuscript is well written and adequately conveys authors’ scientific messages and what authors want to expound. This review bears a value for general readership. I have just a few comments relating to current illustrations displayed on this review.

Figure 1 shows several representative immunofluorescence images for aggregated NETs in different regional compartments. According to the relevant citation shown in the description of the text, some of these figures look to share with previous data used in the authors’ recent reports. If so, the authors may wish to include the sentence(s) evidently indicating that the images were brought from some of those already used in the authors’ previous article(s).

Figure 2 shows a graphical summary of pathologies induced by aggregated NET. This illustration is shown in black/white. If there is no special reason for use of only black color to draw this figure, it is better to depict in different ways (with using multi-colors or selection of differently drawn cartoons). In addition, for describing this illustration some details are written in the main text. I suggest the authors to put summarized remarks of the figure in its legend, because the explanation in the caption looks to be too digested and sounds insufficient.

I would like to restate the comments on Figure 3 common to those on Figure 2, in terms of putting captions with digested outlines in the legend, about what this illustration implies.

Author Response

Please see files attached

Reviewer 2 Report

In the article authors tried to review the newest data that addressed to a major problem in the field of neutrophil biology.

Page 1

Line 31

For description of neutrophil extracellular traps, you use abbreviation (NETs), but in the text you write both NETs or NET.

Please write the abbreviation for neutrophil extracellular traps with same manner in the text.

Page 2, Figure 1

Is the figure 1 original?

Page 3, Line 81

Please correct PADI4 to PAD 4. The abbreviation PAD for Peptidyl arginine deiminase is most accepted.

Page 3

Vascular Occlusions in COVID-19

It was previously show that NETs lead to endothelial cell damage that play a key role in NET-driven thrombogenicity of COVID-19 patients.

Please discuss about endothelial damage and NET-driven thrombogenicity in this section.

Page 4

Coronary Occlusions

It’s well documented that Neutrophils are involved in the pathophysiology of STEMI. de novo production and subsequently expression of functional tissue factor (TF) on the NETs play an important role in the pathophysiology of coronary artery occlusion and STEMI. Unfortunately, in this section you do not describes the ability of neutrophil to express TF on the NETs

Page 5

line 195

The paper by Boeltz, S et al, that describe the role of NETs in severe malaria is just hypothesis article.

The intravascular NET formation in P. falciparum malaria was presented by S. Knackstedt et al in Sci Immunol.

Page 6

Pancreatic Duct

This section in confusing.

As a purpose of this article was to review the role of NETs in occlusions, it is unclear from this section if the authors mean that primary role of NETs in pancreatitis is occlusion of the pancreatic duct. It well known that duct occlusion may cause pancreatitis, but the primary role of NETs in pancreatitis is not duct occlusion. Isn't it?

Pleases rephrase this section.

Author Response

Please see file attached

Round 2

Reviewer 2 Report

All my earlier comments have been sufficiently addressed.